# Research on coupling coordination between construction industry innovation and region economic development in China

**Yong Xiang, Yonghua Chen, Ailing Wan, Yangyang Su** ●*, **Renkai Xiong**

School of Architecture and Civil Engineering, Xihua University, Chengdu, Sichuan, China

* suyangyang@stu.xhu.edu.cn

## Abstract

In numerous developing nations, challenges such as insufficient investment in innovation and limited capabilities for conversion impede the growth of the construction sector, thus affecting the overall economic well-being of these regions. This paper focuses on construction industry innovation (CII) and its correlation with region economic development (RED), providing valuable insights to overcome these challenges and promote sustainable economic advancement. This study references existing literature to devise an evaluation indicator system dedicated for CII and RED. It then proceeds with an empirical analysis of the integration and synergy between CII and the economic development across 31 Chinese provinces from 2012 to 2021. Furthermore, this paper employs ArcGIS and Geoda software to meticulously dissect the spatial distribution characteristics underlying this coordination. The main conclusions are succinctly summarized as follows: CII in China is intricately connected to RED, exhibiting a strong connection that diminishes from south to north. Nonetheless, the coordination level between these factors remains relatively low, with notable regional disparities, particularly from southeast to northwest. The primary obstacles to effective coordination are related to innovation input, output, and economic scale. Additionally, spatial correlation analysis demonstrates pronounced regional clustering, showing stability despite slight fluctuations over the study period. This research underscores the concept of coupling coordination between CII and RED, underpinned by scientific analytical methods. The outcomes provide a definitive guide for advancing the transformation and enhancement of the construction industry while promoting RED.

## 1. Introduction

In most developing countries, the construction sector serves as a cornerstone of the national economy, maintaining a close and interdependent relationship with the coordinated operation of the overarching national economic system [1, 2]. Given the intensifying international competition within the construction field, the integration of innovative technologies into this sector is imperative [3]. CII predominantly revolves around surveying, design, and construction processes, advancing through a holistic amalgamation of technological, organizational,

**Funding:** The author(s) received no specific funding for this work.

**Competing interests:** The authors have declared that no competing interests exist.

personnel, economic, and policy-driven innovations, thereby fostering industrial expansion and national economic development. Despite the embrace of innovation within the construction industry since the dawn of the 21st century, there has been no marked improvement in project management performance [4]. Simultaneously, the levels of CII in most developing countries significantly lag behind those of developed countries [5, 6]. The primary causes for this disparity include a lower degree of industrial modernization, inadequate technological innovation, limited worker skills, and subpar levels of regulatory informatization.

In response to these multifaceted challenges, several developing countries have embarked on exploring cutting-edge technologies such as Building Information Modeling (BIM), modular construction, and artificial intelligence to alleviate the economic, social, and political pressures confronting the construction sector [7]. For instance, certain Latin American countries have enhanced energy efficiency through the adoption of green building technologies [8], South Africa has pursued sustainability in construction via biomimetic approaches [9], and China has reduced both construction time and costs through the implementation of prefabrication techniques [10]. Compared to other developing nations, China's rapidly growing economy has facilitated a more robust advancement in construction innovation. However, according to the 2022 Digital Transformation Report of China's Construction Industry by Yiou Think Tank, construction contributed to 26% of the national GDP in 2020, underscoring its pivotal role in China's economy. Despite this, the construction industry's overall level of digitalization remains relatively primitive, ranking just above agriculture and second to last among major economic sectors [11]. In January 2022, China unveiled the "Fourteenth Five-Year Plan for the Development of the Construction Industry" in an effort to further spur innovation in the field. The plan seeks to improve the industry's levels of industrialization, digitalization, and intelligence in order to facilitate its transformation and upgrade [12]. Within the framework of global sustainable development, the Chinese construction sector must adapt to new economic conditions and industrial structures, transitioning from rapid growth to a more modernized model of economic development [13]. This transition is supported by local policies in provinces and cities such as Beijing, Shanghai, and Guangdong, which continue to realize the economic benefits of innovation in construction [14]. However, regional economic disparities have led to uneven development of construction innovation across different areas. Taking the housing market as an example, in economically prosperous regions, consumer demands have shifted from basic housing needs to higher-quality residences. This shift has compelled construction firms to increase their investment in innovation to meet these evolving market demands [15]. Conversely, in economically weaker regions, the lower demand for advanced housing constrains the necessity for and potential economic benefits of innovation.

Indeed, CII and RED are intricately linked. This dynamic interplay enhances the exchange of technology, capital, talent, and information, fostering mutual and coordinated advancement, and culminating in high-quality systemic growth, as depicted in Fig 1. This process is a critical consideration for China's pursuit of high-quality development and aligns with the United Nations' Sustainable Development Goals [16]. While existing literature has analyzed the relationship between the construction industry and economic development, there remains room for improvement in quantitative analysis and theoretical frameworks. This article offers several potential research contributions: (1) **Empirical Testing of Coupling Coordination:** Empirical Testing of Coupling Coordination: Utilizing an enhanced coupling coordination model, this study quantifies the interplay between CII and RED, employing spatial correlation analysis to unveil the underlying spatial distribution characteristics. This approach provides a novel data perspective, addressing gaps in the existing literature's quantitative analysis. (2) **New Perspectives on Innovation and Regional Development:** As China navigates through a critical phase of high-quality economic advancement, coupled with the ongoing

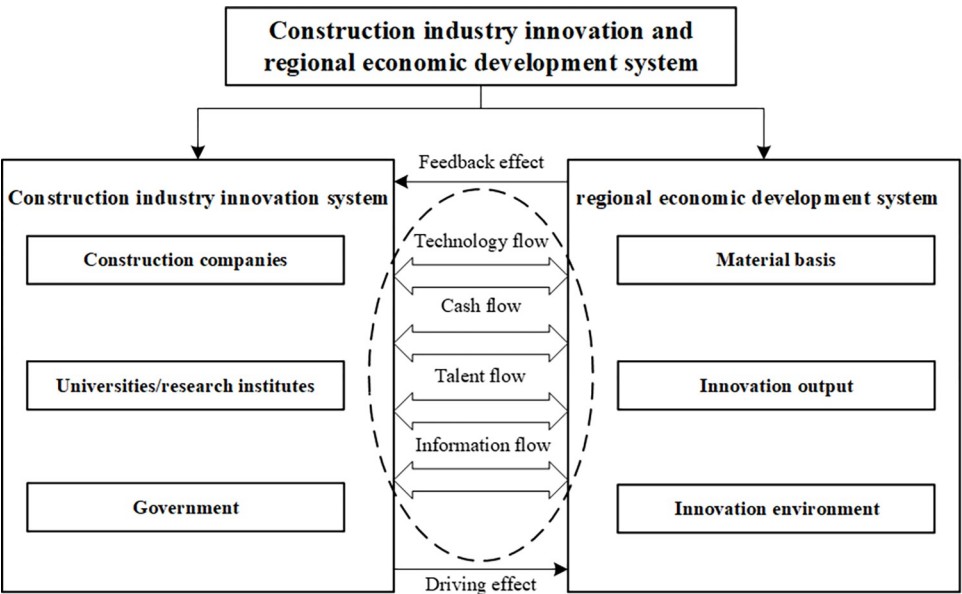

**Fig 1. The theoretical model of coupling coordination CII and RED.**

modernization and enhancement of the construction sector, this study offers a novel perspective on the synergistic development between CII and RED. (3) **Dynamic Insights into Innovation and Regional Development:** This paper aims to objectively depict the dynamic patterns of CII and RED in China. It offers policy recommendations for government and relevant agencies on the direction of transformation in the construction sector and on fostering the coupling coordination between CII and RED.

The subsequent portions of this article are structured as follows. **Section 2:** Summarizes and analyzes pertinent studies on CII, RED, their coordinated interaction, and coordination methods. **Section 3:** Explains the evaluation indicator system, coupling coordination model, and spatial correlation analysis model used for CII and RED. **Section 4:** Explores and discusses the results of the analysis on coupling coordination and spatial correlation in CII and RED, and provide guidance and suggestions. **Section 5:** Offers the conclusion and puts forward prospects and shortcomings.

## 2. Literature review

### Research on CII

Research on CII has its roots in the 1960s when American research institutions and scholars, in response to competitive pressures within the industry, began studying innovation, particularly technological advancements in the construction industry. The primary focus of these studies lies in identifying the sources of innovation drivers, understanding the factors that influence innovation, and exploring the mechanisms for the diffusion of innovation [17–21]. In light of these findings, these studies also propose methods and strategies to bolster the capabilities of CII.

For example, in the quest for more efficient pathways to enhance CII, the majority of studies have employed various research techniques, methods, and models such as external environmental impact models [22], management relationship models [23], maturity models [24], system dynamics models [5, 25], and system classifications methods [26]. These methods help

to deeply analyze and optimize the innovation capabilities of the construction industry. Similarly, in order to improve CII more intuitively, at a macro level, the roles of government, industry, and universities in fostering CII should garner greater attention. Because measuring the efficiency of industry-academia-research collaboration can facilitate the achievement of a continuous and harmonized approach for the spread of innovation [27] and drive the industry towards a transformative phase [5]. At a micro-level perspective, variables including R&D investments, innovation in intellectual assets, a well-educated labor force, and governmental backing impact the vulnerability of innovation capability within the construction industry [28]. Therefore, focusing on the motivating elements that encourage innovation in construction firms can propel CII [29].

However, even though CII has effectively alleviated competitive pressure within the industry, some scholars have become keenly interested in the relationship between CII and economic development and have begun a series of studies [30].

## Research on the interplay between CII and RED

In 1970, Strassmann pioneered the analysis of the correlation between the construction sector and economic development, positing that the value added per capita by the construction industry could enhance the Gross Domestic Product (GDP) per capita [31]. This insight prompted scholars from various nations to delve deeply into the dynamics between the construction industry and economic growth [32–37]. For instance, the construction industry in India holds a central position in the national economic growth. One of its primary contributions is the creation of employment opportunities, which in turn positively impacts the rate of Gross Domestic Product (GDP) expansion and contributes significantly to the overall economic output. This underscores the industry's vital role in fostering economic activity and employment, marking it as a crucial engine for India's economic development [36]. In Malaysia, studies conducted over the periods 1985–1998 and 1998–2009 revealed a significant positive correlation between the efficiency of the construction sector and the national GDP. This correlation indicates that enhancements in productivity within the construction industry effectively propel Malaysia's economic growth, highlighting its vital role in advancing national economic development and overall GDP [37]. In China, the construction industry holds significant backward linkages, exerting a profound impact on the national economy [35]. It maintains a stable output, demonstrating notable causal relationships with the overall macroeconomic scale, the scale of construction industry production, and various national fiscal and monetary policies [34]. These interconnections emphasize the construction sector's importance within the Chinese economy and its influence across a range of economic dimensions and policies.

As economists and business leaders increasingly recognize the pivotal role of efficiency and effectiveness within the global construction sector for economic and social recovery following the global financial crisis [38], there is a growing consensus that technological innovation will yield substantial economic benefits for the construction industry. Studies have shown that by implementing innovative management strategies, the construction industry can achieve economically sustainable development [30]. However, with the advent of digital technologies such as BIM, CII have started to enhance economic benefits through energy-saving and efficiency improvements [39, 40]. These findings confirm CII ability to effectively promote economic development. Yet, in vast developing countries like China, the contributions of innovative elements in the construction industry to overall economic growth often exhibit significant regional disparities [41]. Therefore, it is imperative to explore the relationship between CII and RED within China. However, much scholarly work tends to focus on the unilateral

promotional relationship from CII to RED, with a scarcity of studies on the coordination mechanisms and empirical analysis between them. Coordination refers to the synergistic and harmonious interaction within or between systems [42], making the study of the coordinated impact of CII on RED a topic worthy of in-depth exploration.

## Research on coordination

At present, there are numerous methodologies for researching coordination effects, including system dynamics models [43], grey system theory [44, 45], data envelopment analysis (DEA) [46, 47], and coupling coordination models [48]. While scholars have applied these methods, they have also identified their limitations. For example, the DEA model can evaluate coordination but may fail to provide specific recommendations, requiring further investigation to understand the underlying causes of coordination levels [49]. The grey model GM (1, N) assumes linear relationships between factors, which can lead to biases when many non-linear factors are involved [50]. In contrast, system dynamics models exhibit complex structures and rely on accurate representations of actual systems to deliver reliable results [43]. However, abstract concepts such as social factors and political systems may limit the accuracy of these models.

Studying the coordination between CII and RED requires exploring their compatibility over various periods and identifying the characteristic patterns of their spatiotemporal evolution [51]. The coupling coordination model is particularly suited for these requirements. However, there is no universal model for assessing the coordination degree of system coupling [52]. For instance, Cao Lijun and others have constructed a regional sustainable development coordination model using the ratio evaluation method, which allows for quantitative assessments of the coordination levels among regional population, resources, environment, and economic subsystems [53]. However, this method struggles to accurately judge the coordination degree under different circumstances. Earlier, Liu Yansui and Li Yurui used the elasticity coefficient method to construct a model coupling cultivated land and agricultural labor changes, revealing the spatiotemporal coupling characteristics of these variables [54]. However, this method cannot determine their coordination level. Wang Shujia proposed an improved coupling coordination model that not only makes more accurate judgments about system coordination but also determines the degree of coordination between systems [55]. Therefore, it is extremely appropriate for this article to study the coordination between CII and RED based on this improved coupling coordination model.

In summary, upon reviewing the extensive body of existing research, it becomes evident that literature within the domains of CII and RED primarily explores from a micro-perspective, focusing on their respective application prospects or external influencing factors. However, a macro-level examination of the coordination issues between the two has been conspicuously absent, overlooking the significant interactive coupling effects inherent between CII and RED. In the context of China's pursuit of high-quality development and the United Nations' Sustainable Development Goals, a scientific understanding of the interactive relationship between CII and RED is essential. Recognizing the overall status of coordinated development, its spatiotemporal characteristics, and evolutionary patterns, and uncovering the mechanisms of influence and spatial effects of CII and RED, will provide crucial insights for advancing sustainable development within the construction industry.

## 3. Methodology

### 3.1 Analysis on the construction of indicator evaluation system

**3.1.1 The concept of evaluation index system.** Scholarly literature concerning the evaluation indicators for CII presents two primary methodologies: Firstly, Wang Shoufang [56]

proposes a classification of CII into six dimensions, though challenges arise in quantifying some of these dimensions. Secondly, Li Jia offers a more results-oriented and practical four-dimensional indicator system [57]. Building on these foundations, this paper proposes an evaluation system for CII that encompasses the innovation environment, investment, and outcomes, aiming to enhance both relevance and applicability.

For the evaluation of RED, scholars have traditionally concentrated on aspects of economic growth's sustainability, including economic scale, structure, quality, emissions, energy consumption, employment, and green development [58–61]. This article advances the discourse by refining the RED evaluation metrics to specifically measure the scale, structure, and benefits of economic development. This refined approach allows for a more targeted and effective assessment of economic progress and its sustainable impacts, facilitating clearer insights into the interconnections between economic performance and environmental considerations.

**3.1.2 The principles for constructing an evaluation indicator system.** When constructing an index system to evaluate CII and RED, it is essential to adhere to several guiding principles to ensure the efficacy and accuracy of the evaluation:

1. Coherence: The system should have connected indicators that clearly show how CII and RED are related and how they impact each other. This helps in understanding the overall interaction between CII and economic growth in a region.

2. Accessibility and Analysis: Choose indicators that are simple to collect from data, surveys, or experiments and straightforward for mathematical analysis. They should be simple, clear, and relevant to the goals of the evaluation.

3. Objectivity: The indicators must accurately reflect the true development status of CII and RED. It's important to use reliable data sources and maintain the evaluation unbiased and fair.

4. Comparability: The indicators should allow for comparisons across different regions. This is crucial for analyzing how CII and RED vary from one area to another, helping to draw useful comparisons and conclusions.

**3.1.3 Development of evaluation index system.** Based on data from databases like CNKI, WOS, and other publishing institutions, this paper categorized relevant literature and materials. This classification was extensively used to collect and refine indicators for CII evaluation. Following the purpose and principles behind constructing the evaluation indicator system, we developed a comprehensive system for assessing CII. This system includes 3 primary indicators and 19 secondary indicators, all detailed in Table 1. Similarly, using the same methodology, we constructed an evaluation system for RED. This system also features 3 primary indicators, but with 16 secondary indicators, detailed in Table 2.

## 3.2 Coupling coordination analysis method

The term "coupling" originally comes from physics and refers to the connection between electrical components. It has since been used more broadly to describe how different subsystems interact to form complex systems [72]. In studying systems, "coupling coordination" is used to analyze how different part work together, improving each other's performance in a positive feedback loop [73]. The coordination analysis of CII and RED coupling explores the connections between CII element groups and RED element groups over different time periods. It also looks at how CII systems and RED systems interact with each other and the spatial hierarchy that is involved. However, because the relationship between the two internal elements and

**Table 1. Framework for assessing the level of CII.**

| The objective | The primary indicators | The secondary indicators | Units | Effect direction | Reference |
|---|---|---|---|---|---|
| Indicator system for evaluating CII (A) | Innovation environment (B1) | Evaluating the number of universities through professional assessment (C1) | | + | [29, 56] |
| | | The workforce size in the construction sector (C2) | 10,000 people | + | [56, 62, 63] |
| | | The value added by the construction industry (C3) | 10,000 yuan | + | [56, 62–64] |
| | | Total profits of the construction industry (C4) | 10,000 yuan | + | [56, 57, 62–64] |
| | | Construction project supervision enterprises (C5) | Yuan/ people | + | [29, 56] |
| | | The number of the top 100 most competitive construction enterprises (C6) | | + | [29, 65] |
| | Innovation input (B2) | The number of domestic and international conferences related to CII (C7) | | + | [56, 63] |
| | | The rate of technological equipment in the construction industry (C8) | Yuan/ people | + | [56, 57, 62, 65] |
| | | The number of full-time equivalent (FTE) research and development (R&D) staff in the construction industry (C9) | People year | + | [62, 63] |
| | | Internal expenditure on research and development (R&D) within the construction industry (C10) | 10,000 yuan | + | [57, 63, 65] |
| | | Number of R&D research projects in the construction industry (C11) | | + | [57, 63, 65] |
| | | Intensity of R&D expenditure in the construction industry (C12) | % | + | [48, 57, 63, 65] |
| | | Proportion of senior technical personnel in the construction industry (C13) | % | + | [48, 57, 65] |
| | Innovation output (B3) | Quantity of awards for scientific and technological innovation (e.g., Zhan Tianyou Award) (C14) | | + | [29, 62] |
| | | Number of awards for excellent project management achievements (C15) | | + | [29, 62] |
| | | Number of green building certification projects (C16) | | + | [29, 65] |
| | | Number of technological achievements in the construction industry (C17) | | + | [62, 65] |
| | | Number of patents in the construction industry (C18) | | + | [48, 57, 58, 61, 63, 65] |
| | | Labor productivity in the construction industry (C19) | Yuan/ people | + | [62, 64] |

external factors is uncertain, scientific proof is required to determine whether there is a functional relationship between them. Therefore, this article establishes a suitable coupling coordination model by comparing and selecting the characteristics of multiple coordination methods. The specific steps are as follows (Fig 2):

**3.2.1 Data normalization.** Because there are notable variations in units and orders of magnitude among the original indicators for CII and RED, it is essential to normalize the indicators in order to remove the impact of dimensional units on the calculation results [74].

$$U'_{ij} = \frac{U_{ij} - U_{j\min}}{U_{j\max} - U_{j\min}} \ (positive\ indicators) \tag{1}$$

$$U'_{ij} = \frac{U_{j\max} - U_{ij}}{U_{j\max} - U_{j\min}} \ (negative\ indicators) \tag{2}$$

**Table 2. RED level evaluation indicator framework.**

| The objective | The primary indicators | The secondary indicators | Units | Effect direction | Reference |
|---|---|---|---|---|---|
| Indicator system for evaluating RED (S) | Economic development scale (U1) | Regional gross domestic product (GDP) (V1) | 100 million yuan | + | [48, 58, 66–70] |
| | | Total fixed asset investment (V2) | 100 million yuan | + | [48, 49, 67, 69, 70] |
| | | Industrial value-added (V3) | 100 million yuan | + | [66, 69, 71] |
| | | Retail sales of consumer goods (V4) | 100 million yuan | + | [49, 64, 67, 69, 70] |
| | | Local general public budget revenue (V5) | 100 million yuan | + | [64, 69] |
| | Economic development structure (U2) | Urbanization rate (V6) | % | + | [71] |
| | | Value added by the tertiary industry (V7) | 100 million yuan | + | [49, 67, 70] |
| | | The percentage of the primary industry in GDP (V8) | % | - | [49, 67, 70] |
| | | The dependency ratio (V9) | % | - | [68, 71] |
| | | The proportion of full-time equivalent R&D personnel to the total employed population (V10) | % | + | [58, 71] |
| | | The proportion of R&D expenditure to the regional GDP (V11) | % | + | [58, 61, 71] |
| | Economic development benefits (U3) | Income per capita for urban dwellers (V12) | Yuan | + | [61, 67, 69–71] |
| | | Net income per capita for rural inhabitants (V13) | Yuan | + | [58, 67, 69, 71] |
| | | Per capita Gross Domestic Product (V14) | Yuan | + | [67, 69, 71] |
| | | Total emissions of major pollutants in exhaust gas (V15) | 10000 ton | - | [67, 70, 71] |
| | | Urban registered unemployment rate (V16) | % | - | [49, 67, 68] |

Where, $U_{ij}$ represents the value of an indicator $j$ in a province $i$, $U_{j\min}$ is the maximum value of that indicator among all provinces, $U_{j\max}$ is the minimum value of that indicator among all provinces, and $U'_{ij}$ is the normalized dimensionless value. The normalized value $U'_{ij}$ falls within the range [0, 1].

**3.2.2 Weight determination.** The entropy method utilizes the properties of entropy to evaluate the weights of indicators based on their information content [60]. In this study, this method is employed to enhance the objectivity of the evaluation by representing each

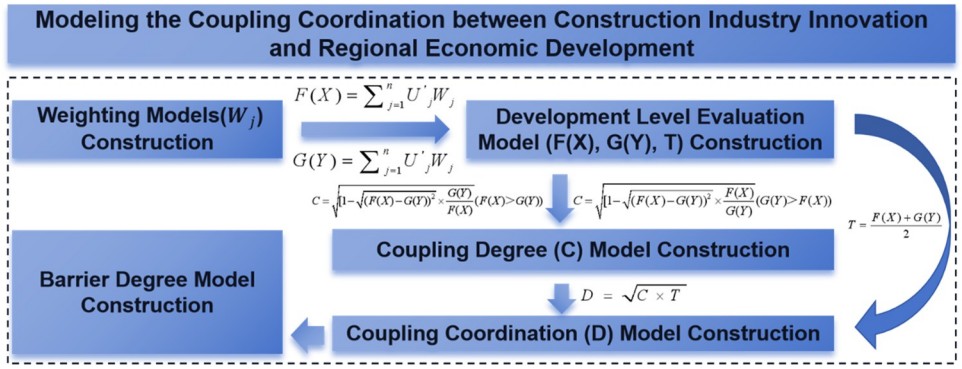

**Fig 2. The model of coupling coordination analysis.**

indicator's contribution to the CII and RED framework [75].

$$P_{ij} = \frac{U'_{ij}}{\sum_{i=1}^{m} U'_{ij}} \qquad (3)$$

$$E_j = -\frac{1}{\ln m} \sum_{i=1}^{m} P_{ij} \ln(P_{ij}) \qquad (4)$$

$$D_j = 1 - E_j \qquad (5)$$

$$W_j = \frac{D_j}{\sum_{j=1}^{n} D_j} \qquad (6)$$

Where, $U'_{ij}$ is the dimensionless value of the indicator $j$ in the province $i$, $P_{ij}$ is the proportion of the ordinal parameter, $m$ is the number of provinces (31 here), $E_j$ is the contribution of the indicator, $0 \leq E_{ij} \leq 1$, $D_j$ is the difference of the indicator $j$, $0 \leq D_{ij} \leq 1$, $W_j$ represents the weight of the indicator $j$, $n$ is the total quantity of indicators.

**3.2.3 The model for evaluating the level of development.** The model for evaluating the levels of CII and RED is demonstrated in Formulas 7 and 8.

$$F(X) = \sum_{j=1}^{n} U'_j W_j \qquad (7)$$

$$G(Y) = \sum_{j=1}^{n} U'_j W_j \qquad (8)$$

Where: $F(X)$ and $G(X)$ individually denote the degree evaluation values of CII and RED, $F(X) \in [0,1]$, $G(Y) \in [0,1]$. $U'_j$ symbolizes the normalized or standardized value of the indicator $j$ under the CII or RED system, and $W_j$ signifies the weight associated with the indicator.

Building on the individual evaluations of the subsystems involved in CII and RED, this study introduces an evaluation value for the overall development level of the integrated system, as shown in Formula 9.

$$T = \alpha F(X) + \beta G(Y) \qquad (9)$$

Where: $T$ signifies the assessment value of the integrated system's development level, $T \in [0,1]$. A greater $T$ value indicates a higher level of development for the integrated system. $\alpha$ and $\beta$ are undetermined coefficients representing the influence of the subsystems on the integrated system, $\alpha + \beta = 1$. While the values of $\alpha$ and $\beta$ have some influence on results but minimally impact the integrated system's overall development trajectory. In this study, both CII and RED significantly influence the system's development and can complement each other to some extent [76, 77]. Therefore, $\alpha = \beta = 0.5$ is chosen, and the above equation is transformed to:

$$T = \frac{F(X) + G(Y)}{2} \qquad (10)$$

**3.2.4 The coupling coordination model.** The coupling degree $C$ is a core component of the coupling coordination model, with values ranging from 0 to 1, representing the strength of the relationship between systems. This paper has established a coupling model specifically for analyzing the interaction between CII and RED in China. Drawing from definitions and

**Table 3. The criterion of coupling degree of CII and RED.**

| Sequence number | Coupling degree (C) | Coupling phase |
|---|---|---|
| 1 | 0 | Uncoupled (UC) |
| 2 | (0, 0.3] | low-level coupling (LC) |
| 3 | (0.3, 0.5] | antagonistic state (AS) |
| 4 | (0.5, 0.8] | running-in state (RS) |
| 5 | (0.8, 1] | high-level coupling (HC) |

methodologies adapted from existing literature, the coupling degree *C* for these two systems has been meticulously defined [55].

$$C = \sqrt{\left[1 - \sqrt{(F(X) - G(Y))^2} \times \frac{F(X)}{G(Y)}\right]} (G(Y) > F(X)) \qquad (11)$$

$$C = \sqrt{\left[1 - \sqrt{(F(X) - G(Y))^2} \times \frac{G(Y)}{F(X)}\right]} (F(X) > G(Y)) \qquad (12)$$

To facilitate analysis, this study divided coupling degrees into five types [2]. Refer to Table 3 for details.

The coupling coefficient is useful for measuring the interaction strength between systems, but it has limitations when it comes to fully describing how well these systems are coordinated. Therefore, this article additionally incorporates the coupling coordination degree *D*, which supersedes the basic coupling degree. The degree of coupling coordination *D* is quantified in Formula 13, offering a more nuanced insight into the overall coordinated development of the system.

$$D = \sqrt{C \times T} \qquad (13)$$

Where, *T* represents the assessment value for the comprehensive development level of the system, and is calculated according to Eq 10. The range of *D* is [0,1], where a higher *D* value indicates a higher degree of coordination within the system. Conversely, a lower *D* value suggests less coordination. When *D* = 1, it signifies the optimal coordination state between the two subsystems.

The assessment standards for coupling coordination used in this study [64, 76] are shown in Table 4.

**3.2.5 Obstacle degree model.** The coupling coordination model is effective for quantitative analysis of the extent of coupling coordination among subsystems. However, it lacks the capability to pinpoint specific factors that constrain the development of coupling and coordination. Therefore, to address this, the obstacle degree model is introduced. This model is designed to compute the obstacle degree for each indicator that limits the system's coordinated development. By doing so, it helps in identifying the core factors that hinder the degree of coupling coordination within the system [78].

1. Determining the type of coupling coordination development
   Before conducting the hindrance diagnosis, it is necessary to ascertain the specific type of lag in the coupling coordination progress for each area. According to the evaluation criteria in Table 3, the hindrance should focus on the subsystem that is lagging.

**Table 4. The criterion of coupling coordination degree of CII and RED.**

| Sequence number | Coupling coordination degree (D) | Coupling doordination level | Comparative relationship | Types of coupled coordinated development |
|---|---|---|---|---|
| 1 | [0, 0.1) | Extremely imbalanced | $F(X)-G(Y)>0.1$ | Extremely imbalanced and lagging economic type |
| | | | $\|F(X)-G(Y)\leq0.1\|$ | Extremely imbalanced and synchrony type |
| | | | $G(Y)-F(X)>0.1$ | Extremely imbalanced and lagging CII type |
| 2 | [0.1, 0.2) | Seriously imbalanced | $F(X)-G(Y)>0.1$ | Seriously imbalanced and lagging economic type |
| | | | $\|F(X)-G(Y)\leq0.1\|$ | Seriously imbalanced and synchrony type |
| | | | $G(Y)-F(X)>0.1$ | Seriously imbalanced and lagging CII type |
| 3 | [0.2, 0.3) | Moderately imbalanced | $F(X)-G(Y)>0.1$ | Moderately imbalanced and lagging economic type |
| | | | $\|F(X)-G(Y)\leq0.1\|$ | Moderately imbalanced and synchrony type |
| | | | $G(Y)-F(X)>0.1$ | Moderately imbalanced and lagging CII type |
| 4 | [0.3, 0.4) | Slightly imbalanced | $F(X)-G(Y)>0.1$ | Slightly Imbalanced and Lagging Economic Type |
| | | | $\|F(X)-G(Y)\leq0.1\|$ | Slightly imbalanced and synchrony type |
| | | | $G(Y)-F(X)>0.1$ | Slightly imbalanced and lagging CII type |
| 5 | [0.4, 0.5) | On the brink of imbalance | $F(X)-G(Y)>0.1$ | On the brink of imbalance and lagging economic type |
| | | | $\|F(X)-G(Y)\leq0.1\|$ | On the brink of imbalance and synchrony type |
| | | | $G(Y)-F(X)>0.1$ | On the brink of imbalance and lagging CII type |
| 6 | [0.5, 0.6) | Barely coordinated | $F(X)-G(Y)>0.1$ | Barely coordinated and lagging economic type |
| | | | $\|F(X)-G(Y)\leq0.1\|$ | Barely coordinated and synchrony type |
| | | | $G(Y)-F(X)>0.1$ | Barely coordinated and lagging CII type |
| 7 | [0.6, 0.7) | Primary coordinated | $F(X)-G(Y)>0.1$ | Primary coordinated and lagging economic type |
| | | | $\|F(X)-G(Y)\leq0.1\|$ | Primary coordinated and synchrony type |
| | | | $G(Y)-F(X)>0.1$ | Primary coordinated and lagging CII type |
| 8 | [0.7, 0.8) | Intermediate coordinated | $F(X)-G(Y)>0.1$ | Intermediate Coordinated and Lagging Economic Type |
| | | | $\|F(X)-G(Y)\leq0.1\|$ | Intermediate coordinated and synchrony type |
| | | | $G(Y)-F(X)>0.1$ | Intermediate coordinated and lagging CII type |
| 9 | [0.8, 0.9) | Good coordinated | $F(X)-G(Y)>0.1$ | Good coordinated and lagging economic type |
| | | | $\|F(X)-G(Y)\leq0.1\|$ | Good coordinated and synchrony type |
| | | | $G(Y)-F(X)>0.1$ | Good coordinated and lagging CII type |
| 10 | [0.9, 1] | Highly coordinated | $F(X)-G(Y)>0.1$ | Highly coordinated and lagging economic type |
| | | | $\|F(X)-G(Y)\leq0.1\|$ | Highly coordinated and synchrony type |
| | | | $G(Y)-F(X)>0.1$ | Highly coordinated and lagging CII type |

2. Calculation of factor contribution

Contribution of factors ($N_j$) represents the weight of a solitary factor to the total objective, as shown in Formula 14.

$$N_j = R_j \times W_j \qquad (14)$$

Where, $R_j$ denotes the weight of the $j$-th indicator dimension, and $W_j$ signifies the weight of the $i$-th indicator within the $j$-th dimension.

3. Index deviation calculation

The degree of index deviation ($O_j$) represents the gap between individual and overall objectives. It is used to express the distinction between the standard value of an individual index

and the ideal maximum value of 1, as shown in Formula 15.

$$O_j = 1 - U'_j \tag{15}$$

Where, $U'_j$ denotes the standardized value of a single index.

4. Calculation of single-index obstacle degree

The single-index obstacle degree ($B_j$) is a metric used to measure how much a single indicator affects the overall objective. A higher single-index obstacle degree indicates that the indicator has a greater impact on achieving the overall objective, as shown in Formula 16.

$$B_j = \frac{O_j \times N_j}{\sum_{j=1}^{n}(O_j \times N_j)} \times 100\% \tag{16}$$

5. Calculation of dimension obstacle degree

Dimensional obstacle degree ($M_{Ij}$) is used to measure the impact of a specific dimension on the overall objective. It is calculated by summing the obstacle degrees of all the indicators within that particular dimension. A higher dimensional obstacle degree indicates a greater impact of that dimension on achieving the overall objective, as shown in Formula 17.

$$M_{Ij} = \sum B_{ij} \tag{17}$$

## 3.3 Spatial correlation analysis

This study conducted spatial correlation analysis to examine the coupling coordination between CII and the RED using ArcGIS 10.8 and Geoda software. Due to the adjacency pattern of the study areas, the choice between a Rook-type spatial matrix or a Queen-type spatial matrix does not significantly affect the calculation results. Therefore, this paper employs the Rook-type matrix for spatial weight matrix calculations.

**3.3.1 Global spatial correlation.** Global spatial autocorrelation is a statistical measure the overall clustering and dispersion of regions and is typically quantified using the Moran's I index, as shown in Formula 18:

$$I = \frac{n\sum_{i=1}^{n}\sum_{j\neq i}^{n}w_{ij}(x_i - \bar{x})}{\sum_{i=1}^{n}\sum_{j\neq i}^{n}w_{ij}\sum_{i=1}^{n}(x_i - \bar{x})} = \frac{\sum_{i=1}^{n}\sum_{j\neq i}^{n}w_{ij}(x_i - \bar{x})(x_j - \bar{x})}{S^2\sum_{i=1}^{n}\sum_{j\neq i}^{n}w_{ij}} \tag{18}$$

Where: $n$ represents the overall count of entities in the study area, with $n = 31$ in this study. $w_{ij}$ is the spatial weight matrix constructed for analysis, $x_j$ symbolizes the attribute values of the $i$-th and $j$-th units. $\bar{x}$ and $S^2$ represent the mean and variance of the observed data.

**3.3.2 Local spatial autocorrelation.**

1. Moran Scatterplot

The Moran scatter plot is a detailed examination for analyzing spatial relationships between neighboring regions [79]. It uses standardized values and depicts four clustering types: H-H (High-High), representing similar and high levels of coupling coordination; L-H (Low-High), showcasing regions with low coordination encircled by higher coordination areas; L-L (Low-Low), indicating consistently low coordination levels; and H-L (High-Low), showcasing higher regional coordination surrounded by lower ones [80].

2. Lisa Clusters

For a quantitative measure within each quadrant, "Univariate Local Moran's I" analysis in Geoda software provides a Lisa cluster examination of coupling coordination's degree between CII and RED across 2012–2021. This aims to focus on regions displaying higher spatial significance.

## 3.4 Study region and data origins

This study focuses on the coupling coordination of CII and RED from 2012 to 2021 in China. The study region primarily includes 31 mainland Chinese regions, with the exclusion of the Hong Kong Special Administrative Region, Macau Special Administrative Region, and Taiwan. The indicator data is sourced from sources including the "China Statistical Yearbook," the "China Science and Technology Statistical Yearbook," the "China Construction Industry Statistical Yearbook," as well as a range of provincial and municipal statistical yearbooks, statistical bulletins released on the platforms of provincial and municipal governments, as well as some authoritative websites. Some of the data has been compiled using the initial data derived from the statistical sources.

This study covers the vast data from 2012 to 2021 across 31 Chinese regions. To manage this, specific points were chosen for analysis: 2014, pivotal for reform goals; 2016, the start of the "13th Five-Year Plan"; and 2020, affected by the COVID-19 pandemic. The study divides the period into t1 (2012–2013), t2 (2014–2015), t3 (2016–2019), and t4 (2020–2021) for analysis of CII and RED coordination.

## 4. Results and discussion

This study utilized a coupling coordination analysis method to evaluate the levels of CII, RED, coupling degrees, coupling coordination degrees, and obstacle degrees across 31 provinces in China over the period from 2012 to 2021. It provides a comprehensive assessment of the interplay between CII and RED in China, offering valuable insights into areas that may require policy adjustments or targeted interventions to enhance regional development strategies.

## 4.1 Coupling coordination degree measurement and analysis

**4.1.1 Coupling analysis.** Using Formulas 1–12, the study calculated the assessment scores for CII, the evaluation scores for RED, and the degree of coupling between them. Then, based on the division of the research period previously outlined, the coupling degrees for the four time periods were linked with spatial analysis units using ArcGIS 10.8. This integration enabled the creation of temporal-spatial distribution maps that visually represent the coupling degree between CII and the RED across the 31 provinces in China from 2012 to 2021, as shown in Fig 3.

The coupling degree between CII and RED in China demonstrates a general decreasing trend from the provinces along the Yangtze River to the surrounding regions, as shown in Fig 3. During the period from 2012 to 2021, this coupling degree exhibited minor fluctuations but displayed a slight upward trend overall. Specifically, the central-southern, eastern, and southwestern regions of China showed most provinces in a state of high-level coupling indicating strong interaction and mutual influence between CII and RED. In these areas, the development of one subsystem significantly affects the other, demonstrating a dynamic interplay. Notably, Chongqing, Jiangxi, Shanghai, Fujian, and Zhejiang have maintained a sustained coordination phase over an extended period. Conversely, Tibet and Hainan have remained in an antagonistic phase, reflecting less favorable interactions between CII and RED. Meanwhile,

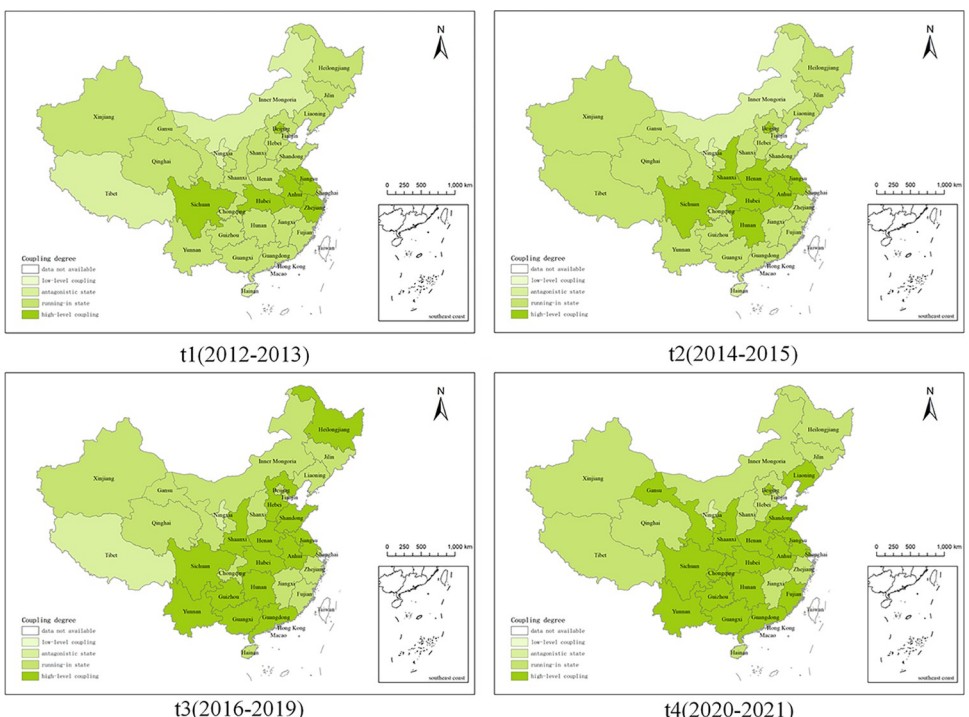

t1(2012-2013)     t2(2014-2015)

t3(2016-2019)     t4(2020-2021)

**Fig 3. The spatiotemporal differentiation of coupling degree. (Source: Drawing approval number GS (2019)1822).**

Guizhou and Yunnan are observed to be transitioning from a coordination phase to a high-level coupling phase. In the northern region, Beijing stands out with a high-level coupling state, while other areas mainly exhibit a coordination phase. Inner Mongolia has shown a notable transition from an antagonistic phase during earlier periods (t1(2012–2013) and t2(2014–2015)) to a coordination phase in later periods (t3(2016–2019) and t4(2020–2021)). The northeastern region predominantly shows a coordination phase among its provinces, with exceptions like Liaoning, which reached a high-level coupling phase by t4(2020–2021), and Heilongjiang during t3(2016–2019). In the northwestern region, Shaanxi maintains a high-level coupling state, contrasting with Ningxia, which is in an antagonistic state, and the rest that have been in a coordination phase for an extended period.

Overall, these results underscore the varied and complex nature of the interactions between CII and RED across China, with certain provinces demonstrating significant fluctuations in their levels of coordination and coupling over time.

**4.1.2 The analysis of coupling coordination degree.** To deepen the understanding and conduct a comprehensive analysis of the interdependent relationship between the CII subsystem and the RED subsystem, this study brings the evaluation values for the level of CII and the level of economic development within regions, and the coupling degree into Formula 13. This formula is utilized to calculate the degree of coordination between CII and economic development at the regional level. The analysis is systematically organized by different time periods and types to showcase how the coordination evolves over time. The results regarding the degree of coordination across various time periods are presented in Figs 4 and 5.

Figs 4 and 5 offer insights into the trends and dynamics of the coordination between CII and RED in China over the period from 2012 to 2021. The coordination degree falls within the range of [0.431, 0.468], showing a consistently increasing trend. This progression highlights a shift in the coupling coordination type from the "brink of imbalance and lagging CII type" to

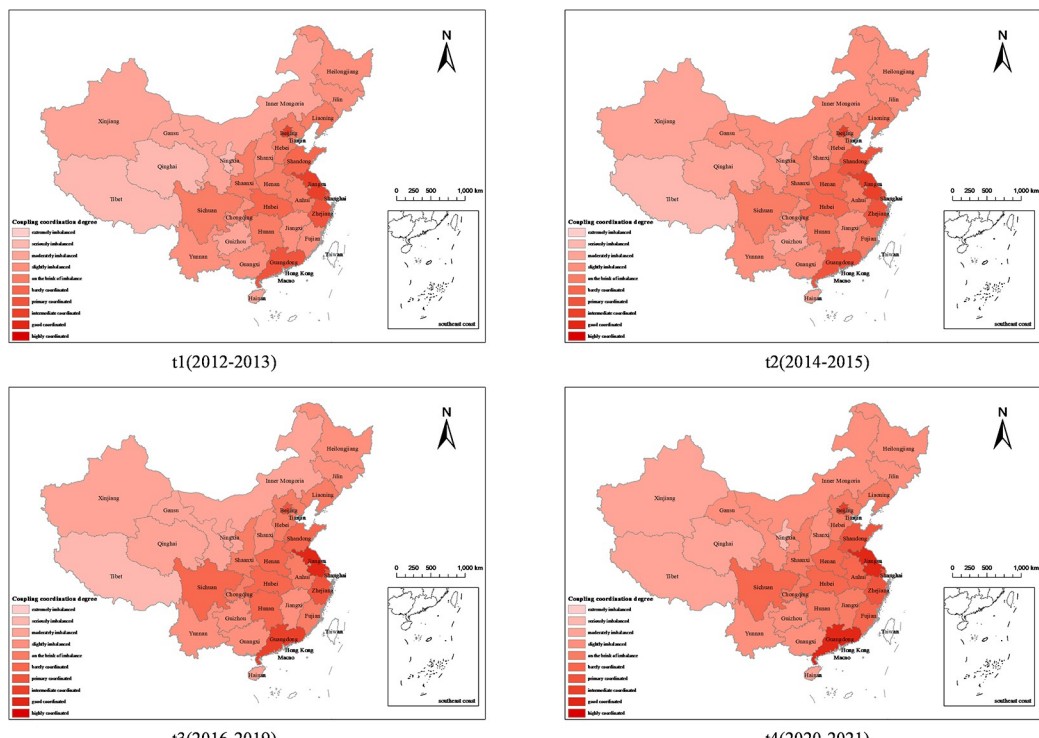

**Fig 4. The spatiotemporal differentiation of coupling coordination degree. (Source: Drawing approval number GS (2019)1822).**

the "brink of imbalance and synchrony type," suggesting a growing mutual influence between CII and RED, though not yet mutually exclusive and still gradually aligning. The relatively low degree of coupling coordination degree can primarily be attributed to the still weak influence of CII on RED, Regional disparities in coupling coordination levels are evident across China:

North China, East China, and Central South regions exhibit coupling coordination levels above the national average. Specifically, East China has shown significant improvement, evolving from "barely coordinated" to "elementary coordination." Central South China has progressed from "borderline disordered" to "barely coordinated." In contrast, North China has seen stable but consistently "borderline disordered" coordination.

Northeast China is experiencing a slight continuous decline, maintaining a state of "mild disarray." Southwest China demonstrates a modest upward trend, transitioning from "mild disarray" to "borderline disarray," primarily characterized by "synchronous" Coordination. Northwest China has the lowest coupling coordination levels, moving from "severely disordered and synchronous" to "mildly disordered and synchronous."

The regions with high coupling coordination are primarily concentrated in China's economically vibrant areas such as the Pearl River Delta, Yangtze River Delta, and Bohai Rim. Jiangsu, Zhejiang, Guangdong, Beijing, and Shanghai are highlighted as having the highest coupling coordination. Conversely, Tibet consistently maintains the lowest coupling coordination, remaining in a state of "severe disarray."

Notably, some regions have experienced changes in their coupling coordination levels, with certain areas experiencing downgrades while others have seen upgrades, reflecting the dynamic nature of economic and industrial development across different parts of China. This detailed regional analysis helps in identifying specific areas where targeted interventions and

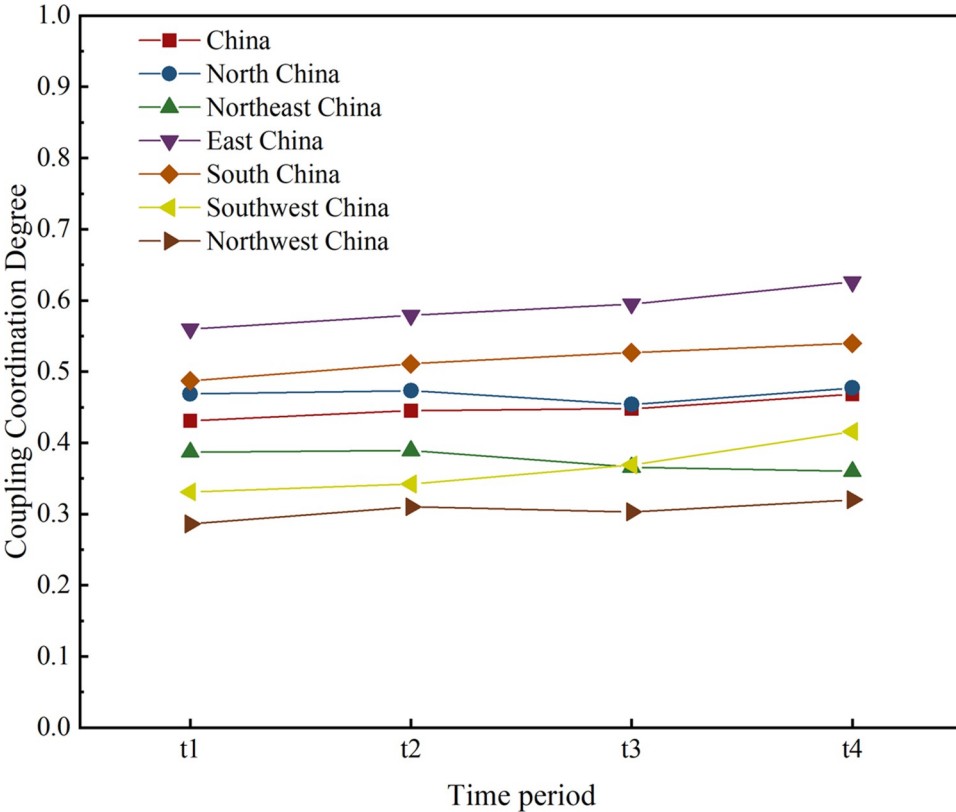

**Fig 5. Temporal segmentation of coupling coordination degree between CII and RED in China and its six economic regions.**

policies might be required to enhance the integration of CII and RED for sustainable economic growth.

**4.1.3 Obstacle degree diagnosis and analysis.** Utilizing using Formulas 14–17, this study calculated the obstacle degrees for various indicators across the 31 provinces in China for each time from 2012 to 2021. This methodology enabled the calculation of average obstacle degrees for each subsystem involved in the study (as shown in Figs 6 and 7).

From Fig 6, the primary obstacle factors for each subsystem of CII are as follows: (1) In the innovation environment, the workforce size in the construction sector and the total profit of

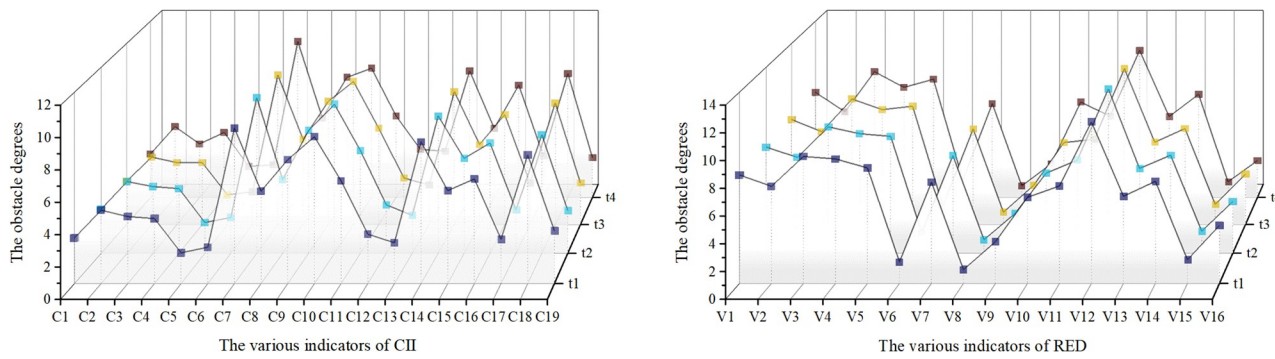

**Fig 6. Obstacle degrees of various indicators for 31 provinces in different time periods.**

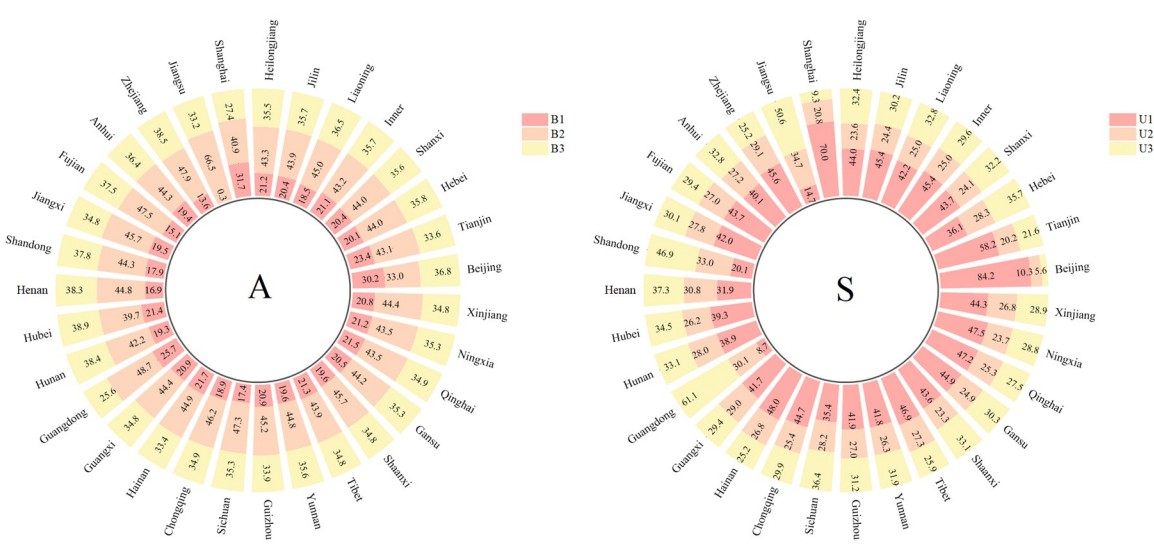

**Fig 7. Average obstacle degree by province and subsystem.**

the construction industry; (2) In innovation input, the number of domestic and international conferences in the construction industry and intensity of R&D expenditure in the construction industry; (3) In innovation output, the quantity of scientific and technological innovation awards (e.g., Zhan Tianyou Award) and the number of patents in the construction industry.

For the RED subsystem, the main obstacle factors are as follows: (1) In the economic development scale, industrial value-added and local general public budget revenue. (2) In the economic development structure, value-added by the tertiary industry and the proportion of R&D expenditure to the regional GDP. (3) In economic development benefits, income per capita for urban dwellers and per capita GDP.

These factors pinpoint specific areas where improvements can greatly enhance the interaction and mutual supportiveness of CII and RED, thereby driving more integrated and sustainable development across the regions. Addressing these obstacles will require focused policies and investments to reduce these impediments and promote more effective coupling and coordination.

Fig 7 provides a detailed ranking of the obstacle levels for the CII and RED subsystems, indicating the primary hurdles in innovation input for CII and economic development scale for RED. This focus is consistent with China's emphasis on boosting technological innovation and comprehensive economic development. By comparing data from Figs 4 and 7, it becomes evident that the 31 regions across China exhibit different obstacle patterns. For instance: Guangdong and other regions show a "CII lag," while Beijing, Shanghai, and others demonstrate multiple obstacles in CII. Some regions depict a "synchronized" state due to obstacles in both CII and RED. These varying patterns across different regions illustrate the diverse obstacle influences on the coupling coordination between CII and RED. Such insights can guide tailored regional strategies to effectively address specific barriers, thereby enhancing overall development.

## 4.2 Spatial correlation analysis

**4.2.1 Global spatial correlation analysis.** In this study, ArcGIS 10.8 and Geoda software were utilized for data preparation and constructing the spatial weight matrix. Subsequently,

**Table 5. Global Moran's I index of coupling coordination.**

| Time | Moran's I | z-value | p-value | Significance |
|---|---|---|---|---|
| 2012 | 0.255 | 2.652 | 0.004 | Yes |
| 2013 | 0.289 | 2.964 | 0.002 | Yes |
| 2014 | 0.282 | 2.902 | 0.002 | Yes |
| 2015 | 0.307 | 3.130 | 0.001 | Yes |
| 2016 | 0.397 | 3.992 | 0.000 | Yes |
| 2017 | 0.335 | 3.386 | 0.000 | Yes |
| 2018 | 0.291 | 2.997 | 0.001 | Yes |
| 2019 | 0.293 | 3.010 | 0.001 | Yes |
| 2020 | 0.248 | 2.602 | 0.005 | Yes |
| 2021 | 0.269 | 2.786 | 0.003 | Yes |

the Moran's I index, which measures the spatial autocorrelation of the coupling coordination degree between CII and RED from 2012 to 2021, was computed using Stata 18 software. The methodology for this calculation adheres to Formula 18, and the results are presented in Table 5.

The results from Table 5 indicate that the Moran's I index for the coupling coordination degree of CII and RED from 2012 to 2021 falls within the range of [0.248, 0.397]. The $z$-values range from [2.602, 3.992], and they are consistently positive. Since the $z$-values exceed the critical value ($z_{terminate}$ = 1.96) for a significance level of $p$ = 0.05 At a significance level of $\alpha$ = 0.05, the $p$-values consistently fall within the range of [0.000, 0.005], all of which are less than 0.05. This suggests that the research units are significantly clustered in space, and there is a positive spatial autocorrelation relationship.

The trend of the Moran's I index trend from 2012 to 2021 indicates a consistently increasing level of coordination between CII and RED. This increase suggests that the interdependencies and synergies between these two sectors are strengthening over time, enhancing the overall economic development landscape. The peak value observed in 2016 signifies the highest level of spatial dependence and clustering among the 31 regions during this period. This peak signifies that during this year, regions with similar levels of coupling coordination between CII and RED were geographically closer to each other, indicating a strong regional alignment in terms of innovation and economic growth strategies.

**4.2.2 Local spatial correlation analysis.** (1) Moran scatter plot analysis

This article utilized Geoda software to create Moran scatter plots from 2012–2021, showcasing the coupling coordination between CII and RED. To examine specific units in various clustering types, unit names were extracted from the plot and organized them on the right side of the scatter plot in a quadrant format. Fig 8 displays these Moran scatter plots for each region's coordination across four time periods, allowing for a clear visual comparison of how regional performance has evolved.

Fig 8 provides a comprehensive visualization of regional clusters in different quadrants based on coupling coordination between CII and RED. Each quadrant represents a specific pattern of interaction between regions and their neighbors:

In the first quadrant, Zhejiang, Jiangsu, Shanghai, Tianjin, Beijing, Anhui, Henan, Hubei, Hunan, and Shandong consistently exhibit H-H clustering, showing high levels of coordination both within themselves and with adjacent areas. Fujian and Chongqing have moved into H-H clustering in later periods, reflecting their enhanced interactions and mutual effects with neighboring regions, with Chongqing notably showing a rising trend in coordination.

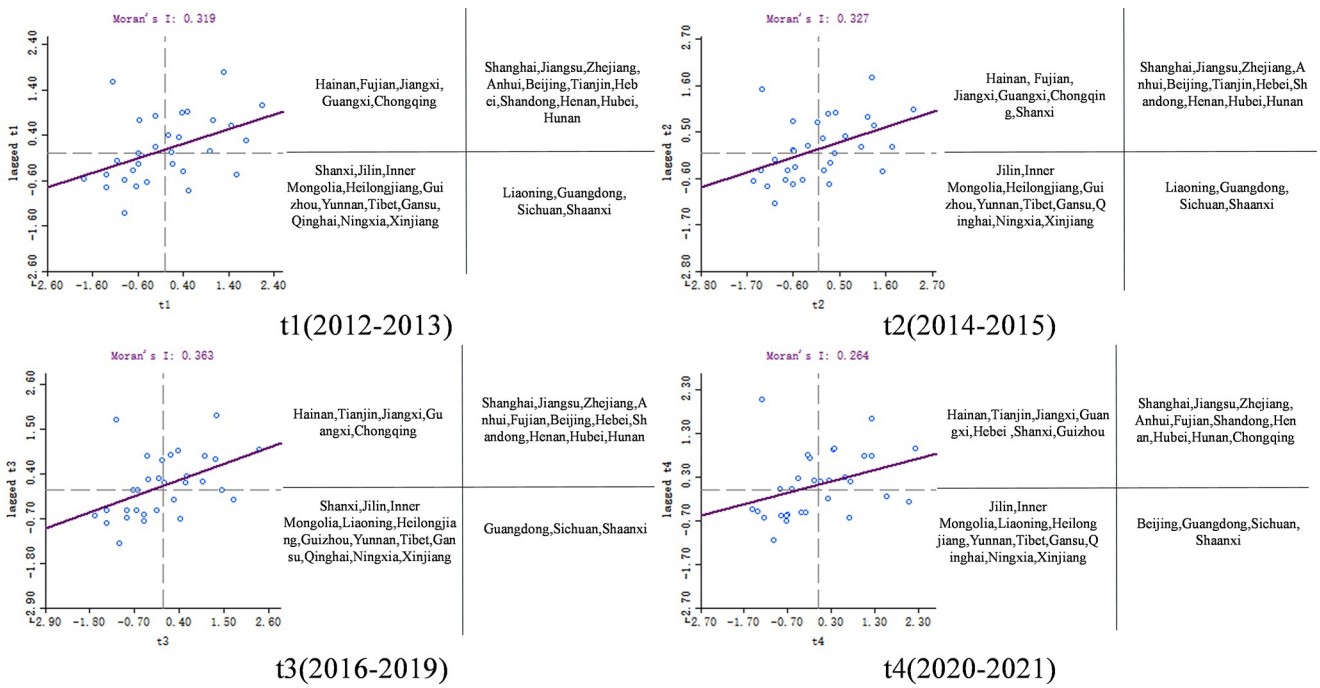

**Fig 8. The spatial and temporal variation in the degree of coupling.**

The second quadrant comprises regions like Guangxi, Jiangxi, and Hainan, which demonstrate L-H clustering. These areas have lower coordination levels internally when compared to their neighboring areas, which perform better. Tianjin and Hebei have transitioned into L-H clustering due to experiencing relatively lower coordination than their better-performing adjacent regions.

The third quadrant involves Jilin, Yunnan, Heilongjiang, and other regions consistently in L-L status, denoting ongoing lower coordination with both within themselves and with their neighboring regions. Liaoning shifted to L-L due to a decline in its coordination levels and similarly lower coordination in adjacent areas.

Sichuan, Shaanxi, and Guangdong consistently display H-L clustering in the fourth quadrant, where these regions maintain higher levels of coordination compared to less coordinated adjacent regions. Beijing notably shifted from H-H to H-L clustering during the study period, indicative of it increasing its coordination level relative to surrounding regions that may not have kept pace.

The study thoroughly analyzed statistical data for regions in each quadrant to understand the dynamics and changes over the study period, with the findings summarized in Table 6.

Table 6 provides insightful observations on the spatial distribution and prevalence of clustering types within China's CII and RED from the data collected. The table reveals a significant

**Table 6. Statistics of the number of the four agglomeration types.**

| Period | H-H number | L-H number | L-L number | H-L number | H-H and L-L ratio | H-L and L-H ratio |
|---|---|---|---|---|---|---|
| t1(2012–2013) | 11 | 5 | 11 | 4 | 71% | 29% |
| t2(2014–2015) | 11 | 6 | 10 | 4 | 68% | 32% |
| t3(2016–2019) | 11 | 5 | 12 | 3 | 74% | 26% |
| t4(2020–2021) | 10 | 7 | 10 | 4 | 65% | 35% |

**Table 7. Lisa aggregate result statistics.**

| Time period | H-H | L-H | L-L | H-L |
|---|---|---|---|---|
| t1(2012–2013) | Zhejiang, Jiangsu, Shanghai, Anhui | Jiangxi | Xinjiang, Qinghai, Gansu | Sichuan |
| t2(2014–2015) | Zhejiang, Jiangsu, Shandong, Anhui, Shanghai | Jiangxi | Xinjiang, Qinghai | Sichuan |
| t3(2016–2019) | Zhejiang, Jiangsu, Shandong, Anhui, Shanghai | Hainan, Jiangxi | Xinjiang, Qinghai, Gansu, Inner Mongolia | Sichuan |
| t4(2020–2021) | Zhejiang, Jiangsu, Shandong, Anhui, Shanghai, Fujian | Hainan, Jiangxi | Xinjiang, Gansu, Inner Mongolia | Sichuan |

the prevalence of L-L clustering regions and a relative scarcity of H-L regions. Remarkably, over 65% of regions consistently fall within H-H and L-L clusters, with this dominance peaking at 74% during the t3 period (2016–2019). This strong presence suggests a prevailing trend of local spatial clustering types within China's CII and RED. The rise in regions classified as H-H suggests a potential for the formation of multiple H-H clusters in the future. The coupling coordination trend displays that regions are clustering into groups of strong and weak coupling, spatially separated. This polarization shows that eastern areas of China tend to exhibit high-high clustering, where both innovation and economic development are strong. In contrast, western regions often show low-low clustering, indicating weaker performance in both sectors. This spatial disparity reflects a "Matthew Effect," where the "rich get richer and the poor get poorer."

(2) Lisa cluster analysis

Moran scatter plots are useful for qualitatively analyzing regional relationships by visually representing spatial autocorrelation, but they do not provide the level of quantitative detail that might be necessary for more in-depth analyses. To address this, this study used GeoDa software to conduct a LISA cluster analysis on CII and RED coupling coordination from 2012 to 2021 that exhibit statistically significant local clustering. The results are summarized in Table 7.

Table 7 displays significant spatial clustering patterns of coupling coordination between CII and RED across various regions in China. The table reveals strong positive correlations in several provinces, including Shanghai, Jiangsu, Zhejiang, Anhui, Jiangxi, Xinjiang, and Sichuan, exhibit notable clustering patterns. Shanghai, Jiangsu, Zhejiang, and Anhui consistently display H-H clustering, while Jiangxi consistently demonstrates L-H clustering. Xinjiang and Qinghai consistently show L-L clustering, and Sichuan remains an H-L clustering center. Shandong transitioned to an H-H center in 2014–2015, influenced by Jiangsu and Anhui's radiating effect. Hainan transformed from non-significant to an L-H clustering center in 2016–2019, expanding the gap with Guangdong. Qinghai, Inner Mongolia, and Gansu show fluctuating clustering patterns, while Sichuan's H-L clustering signifies an ineffective impact on neighboring regions.

## 4.3 Discussion and policy implication

**4.3.1 Discussion.** This study conducts a thorough analysis of the overview, coupling coordination, obstacle diagnosis, and spatial distribution between China's CII and RED. The findings reveal that the current level of coupling coordination between CII and RED in China is generally low and exhibits a slow developmental trend, with significant regional disparities. Given this, exploring ways to enhance the coupling coordination between CII and RED, and reducing the imbalance in coordinated development across regions, has become a critically important issue.

In analyzing the sluggish development of the coupling coordination between China's CII and RED, significant regional disparities emerged as a core factor. Research by Zhang Ruixue and others indicates that innovation in China's construction sector is relatively low and development is unstable [81]. In contrast, the overall regional economies in China show a stable upward trend [82, 83], aligning with the comprehensive assessment results. Therefore, strengthening the development of CII while maintaining a stable level of RED is particularly crucial for improving the current situation. Further analysis reveals considerable disparities in the development levels of CII and RED across Chinese provinces, displaying a regional divergence pattern of "high in the East, low in the West," consistent with the views of Guo [84], Yang [85], and Wei [86]. Although implemented in Eastern China have achieved initial success, they have led to an uneven distribution of resources and innovation capabilities, thus hindering coordinated regional development [87]. Additionally, the concentration of innovative resources and high-tech talent in economically developed areas has resulted in a lack of competitiveness in CII in less developed areas, further hindering RED in these areas [88].

Empirical research indicates that in provinces such as Guizhou and Yunnan, despite achieving good coordination between CII and RED, the low development level of both has not fundamentally resolved developmental challenges. In this regard, some scholars suggest that while focusing on the synergistic development of CII and RED, efforts should also be made to elevate both aspects simultaneously, leveraging technological innovation to drive and support economic development for high-quality coordinated growth [88]. However, in most provinces of the southwest and northwest regions, not only are the levels of CII and RED low, but their coordinated development also faces significant constraints. For these areas, it is recommended to encourage regional cooperation, actively establish platforms for collaborative innovation and open sharing [89]. Additionally, introducing excellent developers from other regions, enhancing the professional level of construction practitioners, and fully utilizing local resources, especially tourism resources, can help create a series of high-quality construction projects. These measures can not only increase the technological content and economic value of construction products but also promote innovation in the construction industry, thereby driving the development of the regional economy.

In the economically developed central and eastern regions such as Beijing, Jiangsu, Shanghai, Zhejiang, Shandong, Guangdong, and Hubei, where the economic and CII environments are relatively advanced, the level of coupling coordination is already high. Therefore, these provinces are not suited to the development strategies used elsewhere. These areas have achieved a certain level of coordination through macroeconomic controls, hence targeted measures are now required. According to the results of obstacle diagnosis, investment and output in construction industry innovation are the main barriers in these provinces [63]. Beijing, Shanghai, and Hubei should further expand their economic scale, converting scale advantages into innovation outcomes to enhance the conversion rate of construction innovation results and to create a more open environment for autonomous innovation in the construction industry [90]. Meanwhile, Guangdong, Shandong, and Zhejiang should actively establish mechanisms that tightly integrate research, production, and the market within the construction sector. Establishing a comprehensive construction industry information system that integrates enterprises, universities, and other innovative entities into the market operations will enhance the continuity and efficiency of the entire construction industry innovation chain.

For those provinces and cities with a foundational base in CII and economic development but a significant disparity in coupling coordination levels compared to areas like Beijing and Shanghai (such as Tianjin, Hebei, Liaoning, Anhui, Fujian, Henan, Hunan, Chongqing, Sichuan, and Shaanxi), geographic location and historical background have limited their autonomous innovation capabilities, and economic development has provided limited impetus for

innovation in construction. These regions should be supported in strengthening inter-regional cooperation and communication, primarily adopting imitation innovation strategies [91]. Taking Fujian Province as an example, the local construction enterprises generally have small scales, poor profitability, and weak overall competitiveness, especially lacking in technical human resources when undertaking large and complex projects, with relatively simplistic operational models. Fujian could look to neighboring Guangdong Province for inspiration, supporting local enterprises in exploring integrated construction and investment operation models, which would facilitate the transformation from traditional construction firms to comprehensive urban development service operators. Additionally, enhancing the training of quality personnel and senior management in the construction industry and improving the competitive edge of local construction enterprises in project bidding are crucial for driving innovation and overall development in the province's construction sector [92].

In an effort to further reduce regional developmental disparities, this study explores the spatial agglomeration characteristics of the coupling coordination degree between China's CII and RED. The research identified several provinces and cities that have established stable agglomeration centers. Although Hong's research, which focused on the Yangtze River Economic Belt, displayed similar phenomena, the limited scope of that study resulted in a lack of generalizability in its conclusions [93]. Our spatial correlation analysis reveals that Jiangsu, Anhui, and Zhejiang have formed stable high-high (H-H) agglomeration centers in the East China region, significantly driving the development of neighboring areas, consistent with the findings of Fang and others [94]. To this end, it is recommended to accelerate the construction of dual-regional linkage mechanisms between "Jiangsu-Anhui" and "Jiangsu-Zhejiang." This could be achieved by establishing platforms for cooperation in scientific achievements and technological transactions, which would facilitate the exchange of information, talent, and technology between regions, and provide a green channel for the construction markets between the two regions, establishing a mutually beneficial government subsidy mechanism [95]. The "Jiangsu-Anhui" linkage mechanism could enhance the coupling coordination development level of Jiangxi, Henan, and Hubei; similarly, the "Jiangsu-Zhejiang" mechanism could boost the coupling coordination development of Fujian, Shandong, and Anhui, further fostering a virtuous cycle. In regions like the southwest and northwest, including Chongqing, Sichuan, and Shaanxi, although the level of coupling coordination development is relatively high, it still ranks at a medium or slightly below medium level nationally, with a clear gap compared to the more developed eastern regions. To effectively aggregate and utilize resources, the western regions should enhance their overall coordination level through synchronized development among Chongqing, Sichuan, and Shaanxi [96].

For provinces like Jiangxi, located within major economic zones or belts but often in a low-high (L-H) agglomeration state, there is a significant disparity in coupling coordination compared to regions like Shanghai, Jiangsu, Zhejiang, and Shandong. In strategic planning, these provinces should focus on integrating with major regional development strategies such as the Belt and Road Initiative and the Yangtze River Delta urban agglomeration, enhancing the attractiveness and absorption of policies. By fully leveraging the impetus of policies like the Belt and Road, they can bridge the gap with the three major economic circles, actively utilize the advantages of surrounding areas with high coupling coordination, strengthen policy connections with these areas, amplify the agglomeration and diffusion effects, and create a broader radiating and pulling impact [97].

Overall, the coupling coordination between China's CII and RED is gradually improving, aligning with the theory of regional economic convergence [98]. Additionally, the results of spatial agglomeration are consistent with innovation diffusion theory [99], especially where initial disparities in CII capabilities exist between regions. However, the dynamic,

comprehensive regional view provided by this study reveals deficiencies in CII capabilities in certain areas and a failure to fully capitalize on local economic strengths. Therefore, for policy-makers, addressing these regional disparities and promoting balanced development will facilitate sustainable and inclusive economic growth. These strategies can not only narrow the developmental gaps between regions but also provide new impetus for overall economic growth.

**4.3.2 Policy implication.**   Enhancing CII and RED levels and their coupling coordination is not only pivotal for advancing the United Nations' Sustainable Development Goals but also a crucial component of China's strategy for high-quality development [100]. Additionally, this process should be accompanied by the construction of robust and inclusive infrastructure to foster sustainable industrialization and stimulate innovation, while also promoting sustained, inclusive, and sustainable economic growth and creating job opportunities. Based on the findings of this study, the following measures are key to achieving these objectives:

Firstly, the research indicates that the coupling coordination between China's CII and RED is generally low and progresses slowly. In response, the government should take the lead in revisiting and revising regulations and policies in the construction industry to ensure these policies fully exert their promotional, regulatory, and motivational effects, thereby driving high-quality development and deep integration of innovation in the construction sector [63]. Moreover, enhancing the science and technology financing system is crucial, as it not only stimulates regional innovation but also facilitates joint advancement in CII and RED [58].

Secondly, the study also finds that the development of coupling coordination between CII and RED varies across Chinese provinces and cities, with distinct barriers in different regions. Therefore, each region should focus on addressing its developmental shortcomings. Governments should formulate policies to guide coordinated development, applying tailored policies based on the specific barriers to coupling coordination in each region, particularly strengthening support for areas with low coupling coordination levels, directing funding towards innovation in the construction sector, and optimizing resource allocation [1]. Additionally, enhancing horizontal cooperation among construction enterprises, implementing project-driven strategies, promoting reform and innovation, and encouraging differentiated innovation models are essential.

Thirdly, considering the spatial agglomeration characteristics where the eastern regions show high-high agglomeration and the western regions low-low agglomeration, to alleviate the polarization in coupling coordination levels, on the one hand, dual-regional linkage mechanisms should be constructed to promote inter-regional cooperation. On the other hand, strengthening the integration of regional planning and expanding the effects of agglomeration and diffusion are necessary to balance regional development and propel the overall economic progress [96].

## 5. Conclusions

This study uses panel data from 31 regions in China from 2012 to 2021to empirically analyze the coordination and coupling between CII and RED. Through the establishment of spatial correlation models, it reveals the spatial distribution features of this coordination. The study deduces several key conclusions:

1. The coupling between CII and RED in China is relatively high but exhibits a diminishing pattern from southern to northern regions, centering around the Yangtze River Economic Belt. CII positively influences RED by expanding market demand, promoting effective marketization of the industry, and generating greater economic and social benefits. Conversely,

RED provides feedback to CII, enhancing it especially through increased government investment in research, which accumulates more funds and talent for CII.

2. The coordination and coupling of CII and RED range between [0.431, 0.468], transitioning from "brink of imbalance and lagging CII type" to "brink of imbalance and synchrony type." Regional disparities are significant, with coordination declining from the southeast to the northwest. High coordination regions are mainly found in the Pearl River Delta, Yangtze River Delta, and the Bohai Rim economic regions. Jiangsu, Zhejiang, Guangdong, Beijing, and Shanghai exhibit the highest coordination. Jiangsu, in particular, achieved a exemplary level of coordination during t3 (2016–2019) and t4 (2020–2021). Conversely, Tibet consistently exhibits the lowest coordination, remaining in a state of severe imbalance. Regions like Ningxia, Gansu, and Inner Mongolia have experienced a downgrading in coordination levels, primarily due to insufficient innovation input and low innovation output rates.

3. The study identifies innovation input, innovation output, and economic development scale as the primary hindrance factors affecting the coordination between CII and RED. The most commonly observed obstacle models in these regions relate to innovation input and output and a combination of–these with economic development scale.

4. The spatial correlation of coupling coordination between CII and RED shows significant clustering, with positive spatial autocorrelation. While slight fluctuations occur, spatial dependency generally stabilizes. A polarization trend is observed where regions with high coordination and those with low coordination tend to cluster spatially. Eastern regions form high-high cluster distribution, while western regions display low-low cluster distributions, a manifestation of the "Matthew Effect."

The findings of this study offer some contributions to both theory and practice in the field of CII and RED. (1) The research results can provide theoretical breakthroughs for understanding the challenges that arise in the interactive development of CII and RED. By analyzing the coupling coordination between these subsystems, the study offers a deeper understanding of the dynamics at play, enriching theoretical frameworks that underpin economic development and innovation strategies; (2) The study proposes optimized paths and policy recommendations aimed at enhancing the coordinated development of the construction industry and regional economies from multiple levels and perspectives. This approach not only aids in strategic planning but also in adapting practices to the unique needs and potentials of different regions; (3) By offering empirical support, the research facilitates the optimization of spatial layouts, which is crucial for effective resource allocation and maximizing the impact of development initiatives. This aspect is particularly valuable for planning authorities looking to balance development across urban and rural areas, or across different regional economies.

By bridging theoretical research with practical applications, this study not only enhances academic understanding but also drives real-world change by informing policy and strategic development. This dual impact ensures that the study's contributions are both valuable and actionable, providing a robust foundation for future initiatives aimed at fostering the integrated growth of the construction industry and regional economies.

This study presents a comprehensive analysis of the coupling coordination between CII and RED. While it offers valuable insights, there are certain limitations and areas for future enhancement:

The multifaceted and dynamic nature of CII necessitates an evaluation across various dimensions, which leads to the development of a comprehensive set of indicators. However, due to the limitations in data availability, some essential indicators crucial for a holistic

evaluation had to be omitted from the system. In their place, more accessible but potentially less specific alternatives were used.

Recognizing this limitation, future research directions should focus on strengthening collaborations with relevant authorities. This enhanced cooperation is aimed at ensuring the continuous and systematic. By doing so, researchers can access a wider range of important indicators, which will significantly improve the quality and accuracy of the CII evaluation system.

Furthermore, research on coupling coordination based on big data is a pivotal area of focus for future studies. Thus, establishing a dedicated information platform to support the analysis of CII and RED interactions is crucial. The complexity and diversity of data sources, along with extended timelines for CII indicators, have previously forced researchers to exclude certain challenging-to-obtain indicators from the evaluation system, potentially leading to gaps in research data. As China continues to prioritize the transformation and enhancement of its construction industry alongside aiming for high-quality RED, the scope of evaluation indicators for both sectors is set to expand. This expansion necessitates a robust approach to the rapid identification, extraction, and analysis of relevant data to thoroughly understand the dynamics between CII and RED.

Consequently, it is essential to progressively establish an extensive networked database that aggregates and updates data regularly from various government and businesses sources. Integrating this data into a unified CII and RED coupling information platform will not only streamline future research efforts but also enhance the precision and relevance of the findings.

## Supporting information

**S1 Data.**
(DOCX)

## Author Contributions

**Conceptualization:** Yong Xiang, Yonghua Chen.

**Data curation:** Renkai Xiong.

**Investigation:** Renkai Xiong.

**Methodology:** Yong Xiang, Ailing Wan, Yangyang Su.

**Project administration:** Yong Xiang.

**Resources:** Yong Xiang.

**Software:** Yonghua Chen, Ailing Wan.

**Supervision:** Yong Xiang, Yangyang Su.

**Validation:** Yong Xiang.

**Visualization:** Yonghua Chen, Ailing Wan.

**Writing – original draft:** Yonghua Chen.

**Writing – review & editing:** Yong Xiang, Yangyang Su.

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
