## [Decision Letter · Decision Letter 0]

9 Apr 2024

PONE-D-24-01506Research on Coupling Coordination between Construction Industry Innovation and Region Economic Development in ChinaPLOS ONE

Dear Dr. Su,

Thank you for submitting your manuscript to PLOS ONE. After careful consideration, we feel that it has merit but does not fully meet PLOS ONE’s publication criteria as it currently stands. Therefore, we invite you to submit a revised version of the manuscript that addresses the points raised during the review process.

We look forward to receiving your revised manuscript.

Kind regards,

Saeed Banihashemi, PhD

Academic Editor

PLOS ONE

Reviewers' comments:

Reviewer's Responses to Questions

**Comments to the Author**

1. Is the manuscript technically sound, and do the data support the conclusions?

Reviewer #1: No

Reviewer #2: Yes

2. Has the statistical analysis been performed appropriately and rigorously? 

Reviewer #1: I Don't Know

Reviewer #2: Yes

3. Have the authors made all data underlying the findings in their manuscript fully available?

Reviewer #1: No

Reviewer #2: Yes

4. Is the manuscript presented in an intelligible fashion and written in standard English?

Reviewer #1: No

Reviewer #2: Yes

5. Review Comments to the Author

**Reviewer #1**: This is an interesting topic that can help/inspire academics and policy makers. However, the current status of the work doesn't worth publishing for English readers. For example, the theoretical model did not receive any explanation and debate in the Lit Rev. The work apparently lost it quality in coherence and cohesiveness.

**Reviewer #2: **1. The description of the literature review should be more logical and readable

2. The author introduced the research background of this paper too much, but they does not explain the realistic background of this research very well

3. The author should summarize the existing research gaps and highlight the innovation of this paper after completing the literature review.

4. The data source should follow the introduction of the method

5. The author needs to introduce the reasons for the research method and combine it with the innovation of the research in this paper.

6. The authors should add maps to analyze spatial and temporal heterogeneity

7. The discussion should focus on describing the differences between the article study and other scholars' studies, thus highlighting the relevance and academic value of the article, the following literature should be helpful for your research: (1)Reduction pathways identification of Agricultural Water Pollution in Hubei Province, China. (2) A differential game of water pollution management in the trans-jurisdictional river basin. (3) Coordination of the Industrial-Ecological Economy in the Yangtze River Economic Belt, China.

8. This article has obtained some interesting findings through the models, but these findings need to be further verified from theory or actual conditions. Also, further highlight the contribution of this article.

9. What are the managerial insights/policy implications of this study?

10. English presentation requires more refinement.

6. PLOS authors have the option to publish the peer review history of their article (what does this mean?). If published, this will include your full peer review and any attached files.

Reviewer #1: No

Reviewer #2: No

---

## [Author Response · Author response to Decision Letter 0]

25 May 2024

I would like to thank you for the thorough and insightful comments on our manuscript titled “Research on Coupling Coordination between Construction Industry Innovation and Region Economic Development in China.” We have carefully considered all the feedback and made the necessary revisions to improve the quality and clarity of our work. Below, we provide detailed responses to each comment and outline the changes made in the revised manuscript.

Reviewer 1:

Comment: This is an interesting topic that can help/inspire academics and policy makers. However, the current status of the work doesn't worth publishing for English readers. For example, the theoretical model did not receive any explanation and debate in the Lit Rev. The work apparently lost it quality in coherence and cohesiveness.

Response:

Thank you for your valuable feedback. We have addressed your concerns by making the following major revisions:

Firstly, we have significantly expanded the Literature Review section to include a comprehensive explanation and debate on the theoretical model used in our study. For example, we added the reasons for choosing this theoretical model, the differences from other similar models, and the specific applications of similar models. Based on this, we strengthened the explanation of the theoretical model to improve the value of this article; Additionally, we removed conceptual introductions to other theoretical models because we considered them redundant compared to the limitations of the models. 

Secondly, we have restructured the manuscript to enhance its coherence and cohesiveness, ensuring that each section logically flows into the next. This involved reorganizing sections to maintain a clear narrative, refining transitions between sections, and ensuring that each part of the manuscript consistently supports our central argument. For example, the introduction now seamlessly leads into the Literature Review, which sets the stage for the Research Methodology. We have also refined the Discussion and Conclusion sections to tie all findings back to our theoretical model, enhancing the overall cohesiveness of the manuscript.

Thirdly, we conducted a thorough language review to improve the clarity and readability of the manuscript for English readers. This included proofreading for grammatical accuracy, improving sentence structure, and simplifying complex language where possible.

We believe these revisions have significantly improved the quality and clarity of our manuscript, making it more suitable for publication.

Reviewer 2:

Comment 1: The description of the literature review should be more logical and readable.

Response:

Thanks to the reviewer's suggestion, we have made significant modifications to the literature review in response to the issues raised about the logic and readability of the literature review, but the general structure has not changed. First, in the CII-related research section, the section on key factors that help improve efficiency and progress was deleted because it has a weak correlation with CII and strengthens its relationship with the next section; Secondly, in the second part, the conceptual definition of RED is deleted, and a research introduction on the interaction between the construction industry and economic development is added, which leads to the introduction that technological innovation will further enhance the contribution of the construction industry to promoting RED; Then, in the third part, the concept of partial coordination methods is deleted, and the advantages of using the research methods in this article to improve the coupled coordination model are emphasized. Through the above modifications, we aim to improve the logic and readability of this literature review.

Comment 2: The author introduced the research background of this paper too much, but they does not explain the realistic background of this research very well.

Response:

Thanks to the reviewer's suggestion. In fact, as the reviewers said, the paper has too many research backgrounds and less realistic backgrounds. To address this problem, we have deleted some research backgrounds and added some international research on CII and economic development. Examples, the current development status of CII in China and the policies implemented by China to promote CII are added to add some realistic background.

Comment 3: The author should summarize the existing research gaps and highlight the innovation of this paper after completing the literature review.

Response:

Thanks to the reviewer's suggestion, we have added the existing research gaps on the coordination research between CII and RED in the last paragraph of the literature review: “a macro-level examination of the coordination issues between the two has been conspicuously absent, overlooking the significant interactive coupling effects inherent between CII and RED. In the context of China's pursuit of high-quality development and the United Nations' Sustainable Development Goals, a scientific understanding of the interactive relationship between CII and RED is essential. Recognizing the overall status of coordinated development, its spatiotemporal characteristics, and evolutionary patterns, and uncovering the mechanisms of influence and spatial effects of CII and RED, will provide crucial insights for advancing sustainable development within the construction industry.”

Comment 4: The data source should follow the introduction of the method.

Response:

Thank you to the reviewer for your suggestion. We followed your suggestion and placed the data sources after the introduction of the research methods. In fact, this did improve the overall logic of the article.

Comment 5: The author needs to introduce the reasons for the research method and combine it with the innovation of the research in this paper.

Response:

Thanks to the reviewer for your suggestion. In response to this suggestion, we have added the reasons for choosing the improved coupling coordination model in the literature review on coordination research, and compared it with other research methods, we can find that the improved coupling coordination model This is in line with the innovative point of this article to reveal the overall situation, spatiotemporal characteristics and evolutionary rules of the coordinated development of CII and RED in China.

Comment 6: The authors should add maps to analyze spatial and temporal heterogeneity

Response:

Thanks to the reviewer's suggestion. In response to this suggestion, this article has re-modified the original Figures 3 and 4. Using arcgis10.8 software, the Chinese geographical map with the review number GS (2019) 1822, and calculation data, we have obtained new 3 and 4, and use this to analyze spatial and temporal heterogeneity.

Comment 7: The discussion should focus on describing the differences between the article study and other scholars' studies, thus highlighting the relevance and academic value of the article, the following literature should be helpful for your research: (1) Reduction pathways identification of Agricultural Water Pollution in Hubei Province, China. (2) A differential game of water pollution management in the trans-jurisdictional river basin. (3) Coordination of the Industrial-Ecological Economy in the Yangtze River Economic Belt, China.

Response:

Thank you for the reviewer's suggestion. This is also true. Our discussion lacks differences from other scholars' research. Therefore, we first carefully read the three references you provided, learned the highlights and features, and discussed the original text. Major modifications. These mainly include: (1) Comparing the level measurement results of CII and RED in the article with other literature, it is found that the results are similar, and the low overall development level of coupling coordination degree also confirms this result; (2) For the obtained the results of the uneven development of regional coupling coordination degree were discussed in detail and compared with other studies. At the same time, corresponding solution measures were obtained with reference to the obstacle degree results; (3) Then by analyzing the spatial correlation research results, it was concluded that China's current the coupling coordination degree of CII and RED is polarized, and some areas show obvious spatial agglomeration. In this regard, we compared other similar academic studies and found that there are certain commonalities with this result, and then focused on the shortcomings of some studies. Here we provide our personal insights and further suggestions for improvement measures for this phenomenon. All in all, we have modified this part to the greatest extent possible to highlight the relevance and academic value of the article by deepening the comparison between the article and other studies.

Comment 8: This article has obtained some interesting findings through the models, but these findings need to be further verified from theory or actual conditions. Also, further highlight the contribution of this article.

Response:

Thanks to the reviewers for their suggestions, we have made some additions to this issue at the end of the discussion. We believe that the results of this study are in line with the core content of regional economic convergence theory and innovation diffusion theory. At the same time, the time range of our study is from 2012 to In 2021, the fluctuations in research results also match policy changes. For example, the research results show that the degree of coupling coordination in many regions has fluctuated since the t3 stage. This is the beginning of China's "13th Five-Year Plan" period. Many regional policies There have been changes, so we believe these findings are somewhat consistent with reality. Furthermore, we further highlight the contributions of this paper in the conclusion section.

Comment 9: What are the managerial insights/policy implications of this study?

Response:

Thanks to the reviewer's suggestion, we have added a policy implication section at the end of the discussion and divided it into three points, including how to solve the low and slow development of coupling coordination between CII and RED in China; how to reduce the imbalance of coupling coordination in various regions, and how to strengthen the spatial agglomeration of China's CII and RED coupling coordination to avoid polarization.

Comment 10: English presentation requires more refinement.

Response:

We tried our best to improve the manuscript’s English presentation and made some changes to the manuscript. These changes will not influence the content and framework of the paper. And here we did not list the changes but marked in red in the revised paper.

---

## [Editor Report · Decision Letter 1]

18 Jul 2024

Research on Coupling Coordination between Construction Industry Innovation and Region Economic Development in China

PONE-D-24-01506R1

Dear Dr. Su,

We’re pleased to inform you that your manuscript has been judged scientifically suitable for publication and will be formally accepted for publication once it meets all outstanding technical requirements.

Kind regards,

Saeed Banihashemi, PhD

Academic Editor

PLOS ONE
---

## [Editor Report · Acceptance letter]

24 Jul 2024

PONE-D-24-01506R1 

PLOS ONE

Dear Dr. Su, 

I'm pleased to inform you that your manuscript has been deemed suitable for publication in PLOS ONE. Congratulations! Your manuscript is now being handed over to our production team.

Kind regards, 

on behalf of

Associate Professor Saeed Banihashemi 

Academic Editor

PLOS ONE